# Efficient Biorefinery Based on Designed Lignocellulosic Substrate for Lactic Acid Production

Ying Wang [1,2,3,*] and Ming Gao [4,*]

1 Department of Biological Science, College of Life Sciences, Sichuan Normal University, Chengdu 610101, China
2 National-Regional Joint Engineering Research Center for Soil Pollution Control and Remediation in South China, Guangdong Key Laboratory of Integrated Agroenvironmental Pollution Control and Management, Institute of Eco-Environmental and Soil Sciences, Guangdong Academy of Sciences, Guangzhou 510650, China
3 Chengdu Environmental Investment Group Co., Ltd., Chengdu 610042, China
4 School of Energy and Environmental Engineering, University of Science and Technology Beijing, Beijing 100083, China
* Correspondence: wangyingcqu@gmail.com (Y.W.); gaoming402@ustb.edu.cn (M.G.); Tel.: +86-(028)-8448-0656 (Y.W.); +86-(010)-6233-2778 (M.G.)

**Abstract:** The current study investigated the feasibility of developing and adopting a few state-of-the-art fermentation techniques to maximize the efficiency of the lignocellulosic waste bioconversion. There have been various efforts towards utilizing the fermentable sugars released from the specific parts of lignocellulose, i.e., cellulose and hemicellulose. However, complete utilization of carbon sources derived from lignocellulosic biomass remains challenging owing to the generated glucose in the presence of $\beta$-glucosidase, which is known as glucose-induced carbon catabolite repression (CCR). To overcome this obstacle, a novel simultaneous saccharification and fermentation (SSF) of lactic acid was designed by using Celluclast 1.5L as a hydrolytic enzyme to optimize the generation and utilization of pentose and hexose. Under the optimal enzyme loading and pH condition, 53.1 g/L optically pure L-lactic acid with a maximum volumetric productivity of 3.65 g/L/h was achieved during the SSF from the brewer's spent grain without any nutrient supplementation. This study demonstrated the potential of lactic acid production from the designed lignocellulosic substrate.

**Keywords:** lactic acid; brewer's spent grain; carbon catabolite repression; simultaneous saccharification and fermentation; enzymatic hydrolysis

## 1. Introduction

Lignocellulosic materials are the most abundant, economical and bio-renewable natural resource on earth. Among the diverse types of lignocelluloses, brewer's spent grain (BSG), the barley malt residue generated from the brewing industry, is a potential lignocellulosic substrate for bioconversion of various added-value chemicals owing to its high carbon content [1]. Along with each 100 L of brewed beer produced, approximately 20 kg of BSG is generated as a by-product [2]. It is estimated that more than 30 million tons of BSG are produced annually worldwide, which is expected to further increase with the blooming micro-brewery market. Most of the BSG are discarded directly into landfills or used as animal feed [3]. The chemical composition of BSG depends on the variety of the barley grains and hops, harvest time, and of course the conditions for brewing [4]. In general, BSG is rich in insoluble fiber, i.e., husk, testa and pericarp, as well as storage protein of endosperm cells, which account for around 70 and 20% of its composition, respectively [2]. The fiber consists mainly of cellulose, hemicellulose (i.e., arabinoxylan) and lignin. Thanks to its high carbohydrate content (up to 50%; *w/w*, dry weight), the BSG is attractive for several biotechnological production processes such as energy and biofuels [5], biogas [6], antibiotics [7], enzymes [8] and some important organic acids [9,10].

Recently, environmental pollution caused by littering of plastics is more than just an unsightly problem. Especially, microplastics are accessible and seriously harmful to aquatic organisms in the oceans, with potential threats to human health [11,12]. Lactic acid, the monomer of polylactic acid, has gained increasing attention since it serves as feedstock for the manufacturing of biodegradable plastics [13]. As one of the naturally occurring organic acids, lactic acid has also been widely applied in food, cosmetic, leather, textile and pharmaceutical industries [14,15]. Global lactic acid demand is expected to reach 1960.1 kilotons by 2020. Lactic acid can be manufactured by both chemical synthesis and biorefinery-based lactic acid fermentation. Compared to chemical synthesis, the biotechnological process offers several advantages including utilization of renewable substrate, low energy requirements, mild production temperature and high purity of final products [16]. Although more favorable than the chemical method, the biorefinery process is economically less viable owing to its high cost of food-based substrate in lactic acid fermentation. Thus, the abundant availability and non-edible characteristics of lignocellulosic biomass make them competitive substrates for sustainable lactic acid production [17].

To date, a few works have reported the potential of lactic acid bacteria-based biorefineries from BSG [18–20]. Generally, supplementation of several nutrients to the fermentation substrates is crucial for the utilization of BSG by lactic acid bacteria [21]. While carbon sources in lignocellulosic hydrolysate are required to generate energy for proliferation, other nutrients such as nitrogen sources, metal ions and vitamins are always supplemented to compensate the nutritional deficiencies [14]. Researchers have supplemented yeast extract [16,20], MRS broth medium components (except the carbon source) [16], malt rootlets extract or soybean meal extract [17], and thin stillage [19] to BSG hydrolysate so as to enhance lactic acid production. Although supplementation of these nutrients seems to be beneficial for lactic acid fermentation with BSG, industrialized biorefineries should be preferable to avoid nutrient supplementation to reduce the overall cost and simplify the substrate composition.

On the other hand, lactic acid fermentation starts from pretreated BSG followed by enzymatic saccharification. Enhanced lactic acid production needs to be achieved by overcoming the incomplete utilization of sugar derived from BSG. However, the hydrolysates of lignocellulosic materials are composed not only of hexoses (such as glucose and cellobiose) but also pentoses (such as xylose and arabinose). Lactic acid bacteria demonstrate utilization of glucose as a preferred sugar over other sugars that might be present and will suppress their catabolism until glucose is fully consumed [22,23]. The carbon catabolite repression (CCR) mechanism is responsible for the bioconversion of carbon sources in a sequential manner, which in turn is expressed as delays in both sugar consumption and final lactic acid generation [23]. In order to achieve efficient lactic acid fermentation from a lignocellulose-derived sugar mixture, we recently established a unique strategy to co-ferment cellobiose and xylose to overcome glucose-induced CCR by *Enterococcus mundtii* in batch fermentation. As a result, 163 g/L lactic acid with a yield of 0.87 g/g consumed sugars was observed by using simulated energy cane hydrolysate [23]. Moreover, this previous work also proved an opportunity for a hydrolysis step without involving exogenous β-glucosidase loading, which would greatly simplify the lignocellulosic biorefinery process and get more cost-effective lactic acid production.

Production of commercial cellulase enzyme depends on its extracellular secretion by common natural habitats of selected microbes such as *Trichoderma reesei* and *Aspergillus niger* [24]. However, most of these enzyme products consist primarily of endo-β-1,4-glucanases (EG, EC 3.2.1.3) and exo-β-1,4-glucanases (or cellobiohydrolases) (CBH, EC 3.2.1.91.), while lacking sufficient activity of β-glucosidases (β-G, EC 3.2.1.21), which play a key role during the final conversion of cellobiose into glucose [25]. The low production of β-glucosidases becomes the major limitation for efficient and complete industrial cellulose hydrolysis [26]. Although numerous studies have succeeded in demonstrating enzymatic hydrolysis of lignocellulosic substrates by supplementing β-glucosidases into commercial cellulase, it involved extra costs and efforts [27,28]. Hence, the aim of this present work was to



establish a novel biorefinery-based lactic acid production from BSG hydrolysate without any nutrient supplementation. The enzymatic hydrolysis was designed by using cellulase without additional β-glucosidases, and CCR in the co-fermentation with generated pentose and hexose therein was evaluated.

## 2. Materials and Methods

### 2.1. Alkaline Pretreatment

Brewer's spent grain was obtained from the Hong Kong Beer Company. The laboratory-scale alkaline pretreatment was performed in a 4-L rotary electric heating digester made by Xian Yang Tong Da Light Industry Equipment Co., Ltd. (Shanxi, China). BSG of 500 g oven dry weight was mixed with 1% sodium hydroxide solution at a 1:10 solid-to-liquid ratio. In the alkaline treatment, the loaded BSG was subjected to a temperature of 121 °C for 15 min. The pretreated BSG was then washed with water until near neutral pH, dried at 50 °C to 50% moisture content, and stored at 4 °C for further enzymatic hydrolysis. The composition of glucan and hemicellulose of the pretreated dry matter were estimated as 18.2% and 20.4%, respectively, based on Laboratory Analytical Procedures (LAPs) standard protocols of the National Renewable Energy Laboratory (NREL) [29,30].

### 2.2. Enzymatic Hydrolysis

Commercial cellulase Celluclast 1.5L and Cellic CTec2 were kindly provided by Novozymes (China) Investment Co. Ltd. These two kinds of commercial cellulase additions have no negative effect on subsequent lactic acid fermentation by *E. mundtii* [31,32]. Cellic CTec 2 has a specific activity of 150 filter paper units (FPU)/mL, while Cellulast 1.5L has a specific activity of 60 FPU/mL. These two cellulase resources were used in the enzymatic hydrolysis separately. Different cellulase dosages were estimated as 5.0, 7.5, 10.0, and 15.0 FPU/g-dry biomass. Presently, few reports have focused on the biorefinery of lignocellulosic biomass with high solid loading (>10% solid content) [33,34]. A high lactic acid concentration would be obtained with high solid loading of BSG substrate, which resulted in lower costs for the subsequent purification process [35]. Therefore, 10% solid content (solid-to-liquid ratio at 1:10) was selected in BSG pretreatment and enzymolysis experiments. Alkaline pretreated BSG slurries at 10% solid content were hydrolyzed using Celluclast 1.5L and Cellic CTec 2 with four different enzyme concentrations after the initial pH was adjusted to 5.0 with 10 M NaOH. 1 M Sodium acetate (NaAc) was added as buffer to maintain an optimum pH of 4.8–5.2 for enzymes function. All setups were put into a constant temperature orbital shaking incubator (HZQ-X100, PeiYing Co., Suzhou, China) at 50 °C for 72 h with an agitation speed of 200 rpm. Samples were taken at different time intervals and analyzed for sugar generation. After the enzymatic hydrolysis, the BSG slurries were centrifuged at 2000× *g* at 4 °C for 30 min to separate the supernatant from solid residues. The supernatants were further filtrated through a 0.45-μm membrane filter to remove the suspended particles. The BSG hydrolysates were prepared for the separate hydrolysis and fermentation processes.

### 2.3. Microorganism and Inoculum Cultivation

*E. mundtii* CGMCC 22227 was used exclusively in this study. The stock of the strain was maintained at −80 °C in vials containing 15% (*v*/*v*) glycerol. For the seed refresh, 1 mL of the glycerol stock was aseptically transferred into 9 mL of modified Man, Rogosa, and Sharpe (mMRS) medium as described previously [23], and then incubated at 43 °C for 24 h. Then the pre-culture was conducted by transferring 10 mL of the refreshed seed to a 300-mL flask with 90 mL mMRS medium and incubating for 8 h at 43 °C. The mMRS medium contains the following components per liter of deionized water: 10 g peptone, 8 g beef extract, 5 g $CH_3COONa \cdot 3H_2O$, 4 g yeast extract, 2 g $K_2HPO_4$, 2 g $C_6H_5O_7(NH_4)_3$, 0.2 g $MgSO_4 \cdot 7H_2O$, 0.05 g $MnSO_4 \cdot 4H_2O$, 1 mL Tween 80, 10 g cellobiose and 10 g xylose. The medium was adjusted to an initial pH of 7.0 using 10 M NaOH prior to autoclaving at 115 °C for 15 min.

*2.4. Lactic Acid Fermentation*

2.4.1. Separate Hydrolysis and Fermentation (SHF)

SHF for lactic acid production was conducted in 100-mL serum bottles containing 50 mL BSG hydrolysates by 5 FPU/g-dry biomass of Celluclast 1.5L or Cellic CTec2 loading without any nutrient supplementation. Hydrolysis steps were performed at 50 °C and 200 rpm for 96 h, initial pHs of the BSG substrates were adjusted to 5.0 and maintained at 4.8–5.2 with 1 M Sodium acetate. Subsequently, fermentation steps were initiated by inoculating 10% (*v*/*v*) of the pre-culture broth into the BSG hydrolyzed substrates [23]. Batch fermentations were carried out at 43 °C with stirring (200 rpm), and the pHs were maintained by adding 30 g/L $CaCO_3$ as a neutralizing agent in the substrates.

2.4.2. Simultaneous Saccharification and Fermentation (SSF)

For SSF tests with serum bottle, the procedures were the same as described for SHF except the pre-culture broth was inoculated at the beginning of the SSF processes. All the SSF cultures were grown at 43 °C and indicated pH with an agitation speed of 200 rpm by adding 30 g/L $CaCO_3$ as a neutralizing agent.

For SSF experiments with jar fermenter, fermentations were carried out in 1-L jar fermenters containing 0.4 L of pretreated BSG slurry. 5 FPU/g-dry biomass of Celluclast 1.5L or Cellic CTec 2 were added into the BSG substrates to initiate pre-hydrolysis. Main cultures were started with inoculating 10% (*v*/*v*) of the pre-culture broth. The batch fermentations were performed with agitation of 200 rpm at 43 °C, equipping an automatic pH control at 7.0 by 10 M ammonium hydroxide addition [23]. Samples were taken at different time intervals and analyzed for sugars and products.

*2.5. Analytical Methods*

The collected samples were centrifuged at $10,000\times g$ for 10 min to remove solids, and the supernatants were filtered through a 0.45 μm filter and determined concentrations of sugars and products by high-performance liquid chromatography (HPLC, Shimadzu; Kyoto, Japan) equipped with an Aminex HPX-87H column (Bio-Rad; Hercules, CA, USA). HPLC analysis was conducted at the column temperature of 60 °C with 5 mM $H_2SO_4$ as a mobile phase at a flow rate of 0.6 mL/min using an injection volume of 50 μL. The lactic acid productivity (g/L/h) was calculated as the ratio of total lactic acid produced (g/L) to the fermentation time (h). Maximum lactic acid productivity (g/L/h) was calculated between each sampling period within the exponential growth phase. The lactic acid yield (g/g) was calculated based on carbohydrate content as the total lactic acid produced (g) per total carbohydrate content (g). Each experiment was performed at least twice to ensure reproducibility. Values were expressed as means ± standard deviation.

## 3. Results and Discussion

*3.1. Effect of Cellulase on Enzymatic Hydrolysis of Pretreated BSG*

The composition and content of cellulase enzymes are considered crucial for enzymatic hydrolysis, which affects sugar generation and bacterial fermentation [36]. Thus, the optimization of enzyme dosages for fermentable sugar yield is one of the most important stages in the development of an efficient and economical lactic acid production strategy. Compared to Cellic CTec2, Celluclast 1.5L contains similar contents of endo-*β*-1,4-glucanases and exo-*β*-1,4-glucanases, however, just 15.0 U/mL of *β*-glucosidase were detected in Celluclast 1.5L, while 2731 U/mL in Cellic CTec2 [37]. There were no reports on optimizing enzymatic hydrolysis with Celluclast 1.5L to generate cellobiose as one of the main products. In this study, the effects of enzymatic saccharification by different Celluclast 1.5L loadings were initially compared with Cellic CTec2.

Figure 1 shows the released sugars during enzymatic saccharification with 5.0, 7.5, 10.0, and 15.0 FPU/g-dry biomass of Celluclast 1.5L and Cellic CTec2, respectively. Generally, sugar concentrations increased with the enzyme loadings except cellobiose. Since the existence of higher *β*-glucosidase in both Cellic CTec2 and 15 FPU/g-dry biomass of Celluclast

1.5L, the generated cellobiose would be further hydrolyzed into 2 glucose molecules. The highest cellobiose concentration of 11.1 g/L was detected at 70 h enzymatic hydrolysis by 15 FPU/g-dry biomass of Celluclast 1.5L. Arabinose had an average concentration of around 1–2 g/L for all enzymatic hydrolysis processes.

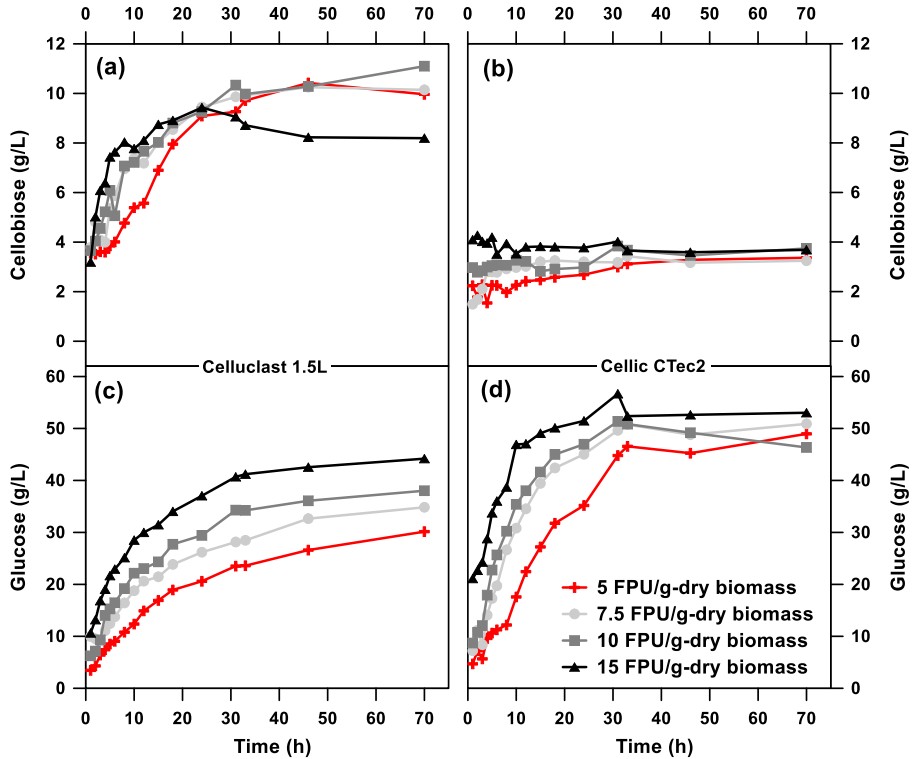

**Figure 1.** Sugar concentrations from enzymatic hydrolysis of pretreated BSG with different loadings of Celluclast 1.5L (**a,c**) and Cellic CTec2 (**b,d**). Data points represent the mean values from three control experiments.

In order to estimate the hydrolysis rate of cellobiose, the parameter $R_{C/G}$ (ratio of cellobiose-to-glucose) was induced to express the concentration ratio of cellobiose to glucose (Figure 2). According to the results, $R_{C/G}$ decreased with both Celluclast 1.5L and Cellic CTec2 loadings. A maximum $R_{C/G}$ up to 0.494 was observed by setting Celluclast 1.5L loading at 5 FPU/g-dry biomass, compared to 0.150 by the same loading of Cellic CTec2. Owing to the low $\beta$-glucosidase activity, the cellobiose was accumulated in the BSG hydrolysate, which made the low enzyme loading of Celluclast 1.5L a preferable candidate for avoiding glucose-induced CCR in the lactic acid fermentation process. It was reported that the lactic acid production cost had increased from US$ 944 to US$ 3692 per metric ton with the increasing price of commercial enzymes [38]. Therefore, the optimized 5 FPU/g-dry biomass Celluclast 1.5L for BSG hydrolysis provided a less expensive way to reduce the overall cost of biorefinery-based lactic acid production. On the other hand, it was noticed that total sugars released by Celluclast 1.5L were slightly lower compared to Cellic CTec2 with the same enzyme loadings. It has been identified that the feedback inhibition by cellodextrins on enzymatic cellulose hydrolysis reactions imposed potential limitations on the efficient lignocellulose utilization [39]. Consequently, a bioconversion process that can reduce generated sugars should be crucial for bypassing the feedback inhibition.

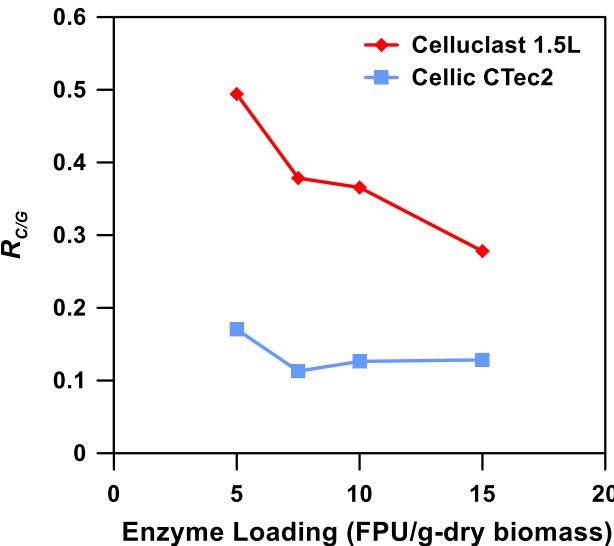

**Figure 2.** Cellobiose-to-glucose ratio, $R_{C/G}$, calculated when 5.0, 7.5, 10.0, 15.0 FPU/g-dry biomass were used in the hydrolysis experiment. Symbols: diamond, Celluclast 1.5L; squares, Cellic CTec2. Data points represent the mean values from three control experiments.

### 3.2. Comparison of SHF and SSF Processes for Lactic Acid Production

The hydrolysis experiment proved that Celluclast 1.5L should be a better option compared with Cellic CTec2 for BSG hydrolysis by our proposed co-fermentation strategy [23]. The formed BSG hydrolysates were further observed in SHF and SSF processes by using *E. mundtii.* SSF means the enzymatic hydrolysis and fermentation are conducted in the same reactor under the same conditions, while SHF refers to processes of enzymatic hydrolysis and fermentation that were conducted separately, each process could thereby be carried out under optimal conditions. Compared with SHF, SSF is usually preferred in industry processes due to lower cost, reduced contamination risk, and lower sugar inhibitory effects. Other advantages of SSF over SHF include the usage of a single reactor for both steps, rapid processing time, reduced feedback inhibition by the generated sugars, and increased productivity [14,40]. However, the requirement for different optimal temperatures and pHs for saccharification and fermentation is the main limiting factor for SSF [14].

In a preliminary fermentation experiment, batch SHF and SSF were performed in serum bottles containing 10% solid content of BSG slurries supplemented with 5 FPU/g-dry biomass of Celluclast 1.5L or Cellic CTec2, respectively. Based on the previous hydrolysis results, the composition of BSG hydrolysate mainly consists of glucose, cellobiose and xylose. As shown in Table 1, higher glucose accumulated at the end of SHF compared to those in SSF processes. For SHF and SSF with Celluclast 1.5L, similar amounts of residual cellobiose (5.99 ± 1.42 and 5.70 ± 1.02 g/L, respectively) were generated. One difference was that cellobiose and xylose were slowly consumed during SSF but almost not in SHF. Due to CCR, bacteria would feed on preferable glucose, instead of cellobiose and xylose [23]. As saccharification and fermentation were simultaneously undergoing in SSF, lower concentrations of sugars were accumulated. Studies showed that CCR can be prevented by retaining glucose less than a certain concentration in the substrate for lactic acid fermentation [14]. In other words, when glucose concentration remains at a relatively low level (i.e., 25 g/L) that does not exceed the threshold of CCR, bacteria would feed on any sugar that was freshly produced from hydrolysis. For Celluclast 1.5L, the lactic acid production of SSF was 25.7 ± 2.27 g/L, while that of SHF was 21.9 ± 3.47 g/L. For Cellic CTec2, the lactic acid production of SSF was 23.1 ± 4.01 g/L, while that of SHF was 20.8 ± 1.62 g/L. Owing to the glucose-induced CCR, a large amount of pentose would not be utilized in the presence of high glucose concentration. Thus, the most efficient composition of hydrolyzed glucan for lactic acid production should keep glucose stay at its disaccharide form, i.e., cellobiose, so that all sugar consumption can be improved more

effectively. In summary, saccharification of pretreated BSG slurries using 5 FPU/g-dry biomass of Celluclast 1.5L in an SSF process enhanced both the utilization of mixed sugars and the production of lactic acid by *E. mundtii*.

**Table 1.** Lactic acid fermentation with pretreated BSG by SHF and SSF processes.

| Fermentation Mode | Hydrolytic Enzyme | $C_{Glc}$ [1] (g/L) | $C_{cel}$ [2] (g/L) | $C_{Xyl}$ [3] (g/L) | $C_{LA}$ [4] (g/L) | $Y_{LA}$ [5] (g/g) |
|---|---|---|---|---|---|---|
| SHF (121 h) | Celluclast 1.5L | 23.0 ± 3.43 | 5.99 ± 1.42 | 10.4 ± 0.88 | 21.9 ± 3.47 | 0.287 |
| SHF (121 h) | Cellic CTec2 | 35.9 ± 4.02 | 3.18 ± 0.33 | 13.0 ± 1.73 | 20.8 ± 1.62 | 0.273 |
| SSF (145 h) | Celluclast 1.5L | 18.2 ± 1.97 | 5.70 ± 1.02 | 7.38 ± 0.79 | 25.7 ± 2.27 | 0.337 |
| SSF (145 h) | Cellic CTec2 | 20.2 ± 0.73 | 4.15 ± 1.83 | 8.96 ± 1.65 | 23.1 ± 4.01 | 0.303 |

[1] Residual glucose concentration; [2] Residual cellobiose concentration; [3] Residual xylose concentration; [4] Maximum lactic acid concentration; [5] Lactic acid yield. Averages with standard deviations are based on three independent fermentations.

### 3.3. Optimal pH for Lactic Acid Production from Pretreated BSG

In addition to the enzyme loading, maintaining optimal operating conditions such as pH is also crucial for the performance of SSF. The typical operating pH for enzymatic hydrolysis ranges between 4.5 and 5.5 [41,42], in contrast to *E. mundtii* exhibited the optimal pH of around 7.0 for lactic acid production [23]. Thus, compromises between the conditions for enzymatic hydrolysis and fermentation are necessary to achieve high sugar utilization, lactic acid accumulation and yield during the SSF process [43]. In an attempt to study the effect of pH on lactic acid fermentation, SSF cultures were conducted under initial pH at 5.5, 6.0, 6.5, 7.0 and 7.5 in serum bottles to verify the optimal pH for lactic acid production.

Table 2 shows the kinetic parameters of SSF at different pH values by using pretreated BSG. Under pH values from 5.5 to 6.0, glucose, cellobiose and xylose consumptions were very slow and almost not utilized until the end of fermentations. Whereas sugar consumption rates were higher from pH 7.0 to 7.5. Although lower residual sugars were observed at pH 7.5, the lactic acid concentration and yield did not exhibit maximum values. After 120 h duration of cultivation, 22.30 ± 3.02 g/L of lactic acid with a yield of 0.292 g/g was achieved at pH 7.0. A small amount of 1.03 ± 0.08 g/L acetic acid was detected as a by-product in this process. Increases in sugar utilization and lactic acid production in SSF of lactic acid with an initial pH of 7.0 have been also reported for other lactic acid producers, including *Bacillus coagulans* [44].

**Table 2.** Effects of pH on SSF of lactic acid from pretreated BSG.

| pH | $C_{Glc}$ [1] (g/L) | $C_{Cel}$ [2] (g/L) | $C_{Xyl}$ [3] (g/L) | $C_{LA}$ [4] (g/L) | $Y_{LA}$ [5] (g/g) |
|---|---|---|---|---|---|
| 5.5 | 38.7 ± 3.13 | 5.64 ± 1.50 | 16.8 ± 2.22 | 5.61 ± 0.53 | 0.074 |
| 6.0 | 29.6 ± 1.37 | 6.86 ± 0.16 | 14.9 ± 1.03 | 13.83 ± 2.18 | 0.181 |
| 6.5 | 22.0 ± 3.28 | 5.87 ± 1.76 | 12.5 ± 1.83 | 18.15 ± 1.79 | 0.238 |
| 7.0 | 16.2 ± 0.93 | 4.32 ± 0.69 | 9.36 ± 0.73 | 22.30 ± 3.02 | 0.292 |
| 7.5 | 14.2 ± 2.34 | 4.27 ± 1.02 | 7.94 ± 0.34 | 19.77 ± 1.28 | 0.259 |

[1] Residual glucose concentration; [2] Residual cellobiose concentration; [3] Residual xylose concentration; [4] Lactic acid concentration at 120 h; [5] Lactic acid yield. Averages with standard deviations are based on three independent fermentations.

### 3.4. Improved SSF of Lactic Acid from Designed BSG Substrates

It was reported that the viscosity of the fermentation medium and initial concentration of fermentable sugars in SSF was associated with the performances of the inoculated bacteria [45,46]. Lower initial sugars would limit cell growth and lactic acid production, whereas higher initial sugars would impose higher osmotic stress [47]. Since a relatively high cellobiose concentration was detected around 20 h of hydrolysis with 5 FPU/g-dry biomass of Celluclast 1.5L, to verify the improved lactic acid production in SSF with prehydrolysis compared with ordinary SSF, 20 h of prehydrolysis prior to the main SSF

was investigated in this study. During the BSG hydrolysis step, pH and temperature were set to the optimum for the hydrolytic enzyme (50 °C, pH 5.0) to ensure the speed of saccharification was maximal, then they were shifted to the optimum of the bacteria (43 °C, pH 7.0) to maximize lactic acid conversion rate during SSF.

The sugar consumption and product concentration of the ordinary SSF and SSF with prehydrolysis processes are presented in Figure 3. Sugars were released and accumulated along with the 20 h of prehydrolysis in SSF, and their consumption was initiated by bacterial inoculation (Figure 3a). The SSF with prehydrolysis exhibited simultaneous consumption of glucose, cellobiose, and xylose during the early phase of fermentation. The released glucose and xylose were depleted in both SSF with prehydrolysis and the ordinary SSF processes, while a low amount of 2–3 g/L of cellobiose was observed and the end of fermentations. Glucose-induced CCR had less effect in this scenario and high sugar utilizations were demonstrated. Correspondingly, the SSF with prehydrolysis could improve L-lactic acid production (maximum concentration, 53.1 ± 2.83 g/L, maximum productivity, 3.65 ± 0.22 g/L/h, and yield, 0.696 g/g) at an optical purity of ≥99.4% compared to those parameters (44.9 ± 1.78 g/L, 3.06 ± 0.40 g/L/h, and 0.588 g/g, respectively) in the ordinary SSF. Both of the two fermentations were homolactic processes since only small amounts of by-product (1–2 g/L acetic acid) were observed. It thus appears that prehydrolysis prior to main SSF can significantly increase the lactic acid production in SSF. This work succeeded in establishing SSF conditions for homolactic fermentation of mixed sugars derived from BSG with low effect of CCR, which have not been previously reported.

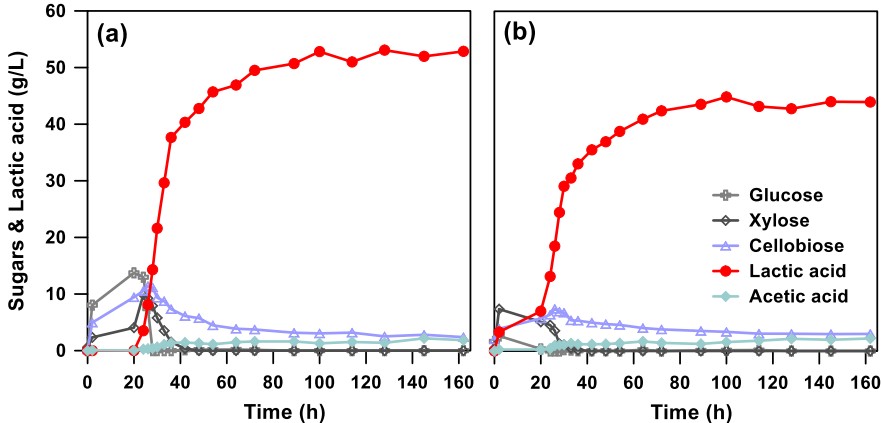

**Figure 3.** Profile of lactic acid fermentation with BSG under conditions of (**a**) SSF with 20 h of prhydrolysis (**b**) ordinary SSF. Data points represent the mean values from three control experiments.

There is currently great interest in efficient SHF or SSF of carbohydrates in hydrolysates derived from lignocellulosic biomasses for lactic acid production. Table 3 summarizes the results of recent reports on lactic acid fermentation by using grain biomass as substrates. Basically, most of these studies investigated lactic acid fermentation from grain-derived lignocellulosic hydrolysates by supplementing the cultures with several nutrients such as MRS broth [17,48], yeast extract [21,49,50] and nitrogen gas [51] for improving the cell growth and lactic acid production. As shown in Figure 4, the yields and productivities of lactic acid vary with different substrates, fermentation modes, as well as producers. Generally, the glucose and starchy substrates usually result in faster lactic acid production [49,50]. To the best of our knowledge, the present study is the first to investigate lactic acid conversion from mixed sugars of glucose-cellobiose-xylose dissolved in lignocellulosic hydrolysate and to minimize glucose-induced CCR under optimized hydrolysis conditions, which led to relatively high productivities of lactic acid and fewer by-products. Our results exposed a great potential for lactic acid bioconversion by using the designed lignocellulosic substrate. Furthermore, a novel biorefinery approach for BSG resource recovery with high-value-added chemical lactic acid production was revealed in this work. The residues

divided from lactic acid fermentation with BSG could be reused as a kind of substrate for composting, which provide a full resource processing technology for BSG valorization.

**Table 3.** Lactic acid fermentation with grain biomass hydrolysates.

| Microorganism | Substrate | Fermentation Mode | Nutrients Supplementation | $C_{LA}$ [1] (g/L) | $Y_{LA}$ [2] (g/g) | $P_{LA}$ [3] (g/L/h) | Ref. |
|---|---|---|---|---|---|---|---|
| *E. mundtii* | BSG | SSF | No | 44.9 | 0.588 | 3.06 | This study |
| | BSG | SSF with prehydrolysis | No | 53.1 | 0.696 | 3.65 | This study |
| *L. delbrueckii* | BSG | SHF | MRS broth | 35.5 | 0.485 | 0.82 | [17] |
| *L. rhamnosus* | BSG | SHF | 50 g/L yeast extract | 39.4 | 0.913 | 1.69 | [21] |
| *L. delbrueckii* UFV H2b20 | BSG | SHF | No | 5.4 | 0.074 | 0.11 | [18] |
| *L. rhamnosus* ATCC 7469 | BSG supplemented with glucose | SHF | 50 g/L yeast extract | 116.1 | 0.933 | 2.0 | [50] |
| *L. plantarum* Δ*ldh1* | Corn stover | SSF | mMRS broth | 21.1 | 0.505 | 0.5 | [48] |
| *L. bifermentans* DSM 20003T | Wheat bran | SHF | No | 62.8 | 0.647 | 1.2 | [52] |
| *L. delbrueckii* IFO 3202 | Rice bran | SSF | $N_2$ gas | 28.0 | 0.483 | 0.78 | [51] |
| *L. sp. MKT-878* | Wheat starch | SHF | 6 g/L yeast extract | 118 | 0.908 | 3.57 | [49] |
| | Wheat starch | SSF with prehydrolysis | 6 g/L yeast extract | 121 | 0.931 | 4.32 | [49] |

[1] Maximum lactic acid concentration; [2] Lactic acid yield based on carbohydrate content; [3] Maximum lactic acid productivity.

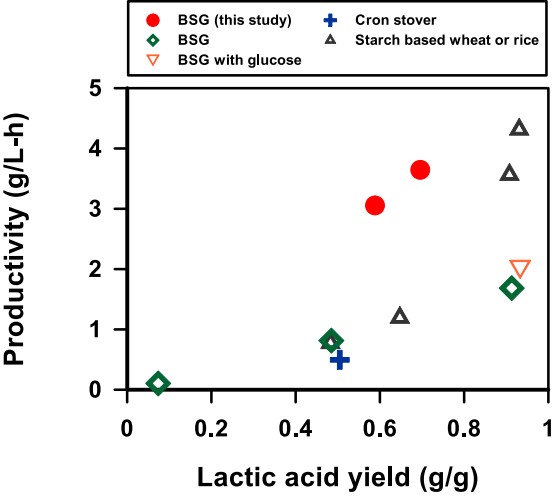

**Figure 4.** Impacts of different grain-derived lignocellulosic hydrolysates and biorefinery processes to yields and productivities of lactic acid. The points show some examples from literatures and this study for comparison.

## 4. Conclusions

This study demonstrated the feasibility of lignocellulosic waste fermentation derived from BSG by using *E. mundtii*. Celluclast 1.5L at an enzyme loading of 5 FPU/g biomass was optimized for BSG hydrolysis to generate pentose and hexose at a preferred $R_{C/G}$. SSF process enhanced the utilization of fermentable sugars in fermentation. Maximum volumetric productivity of 3.65 g/L/h was achieved with 53.1 g/L optically pure L-lactic acid produced by controlling pH at 7.0 during the SSF with 20 h of prehydrolysis. The glucose-induced CCR was successfully minimized in lactic acid production with cellulose

and hemicellulose-derived carbon sources. Lactic acid producing strain could grow in BSG hydrolysate even without any nutrient supplementation might be due to that BSG is not only used as carbon source but also supplied nitrogen source for lactic acid producing strain. Finally, an efficient biorefinery of lactic acid production based on SSF of mixed sugars in BSG hydrolysate was successfully established.

**Author Contributions:** Y.W.: Conceptualization, Methodology, Data curation, Writing—original draft. M.G.: Writing—review & editing, Funding acquisition. All authors have read and agreed to the published version of the manuscript.

**Funding:** This research was funded by the National Key R&D Program of China (Grant NO. 2022YFE0118800), the National Natural Science Foundation of China (Grant NO. 52000133), the Natural Science Foundation of Sichuan Province (2022NSFSC0390), the GDAS' Project of Science and Technology Development (Grant NO. 2019GDASYL-0102005), and "Innovation China" science and technology service program of Jinjiang District (kx002).

**Institutional Review Board Statement:** Not applicable.

**Informed Consent Statement:** Not applicable.

**Data Availability Statement:** Data available on request from the authors.

**Acknowledgments:** The authors thank Novozymes China for providing the Celluclast 1.5L and Cellic CTec2 for experiments.

**Conflicts of Interest:** The authors declare no conflict of interest.

## Abbreviations

CCR: carbon catabolite repression; SSF: saccharification and fermentation; BSG: brewer's spent grain; EG: endo-$\beta$-1,4-glucanases; CBH: exo-$\beta$-1,4-glucanases, cellobiohydrolases; $\beta$-G: $\beta$-glucosidases; LAPs: Laboratory Analytical Procedures; NREL: National Renewable Energy Laboratory; NaAc: sodium acetate; SHF: separate hydrolysis and fermentation; mMRS: modified Man, Rogosa, and Sharpe medium; HPLC: high-performance liquid chromatography; FPU: filter paper unit; $R_{C/G}$: ratio of cellobiose-to-glucose.

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
