# Peer review of "Efficient Biorefinery Based on Designed Lignocellulosic Substrate for Lactic Acid Production"

_fermentation, doi:10.3390/fermentation9080744_

Round 1

Reviewer 1 Report

The work presented by wang and Gao., entitled “Efficient biorefinery based on designed lignocellulosic substate for lactic acid production” describes the production of lactic acid from lignocellulosic substrate after enzymatic hydrolysis The subject is relevant and interesting mostly when cheap carbon sources are used. Overall, the paper is acceptable and it can be published with minor revision.

I have just few comments:

The materials and methods part is sometimes too detailed (§2.5) and sometimes not enough (§2.4).

The figures are too small.

The authors speak about biorefinery in the title but the biorefinery must strive towards no waste. What do they plan to do with the waste from lactic fermentation?

Author Response

Thank you very much for reviewing our manuscript and giving us helpful comments. We made corrections, and we hope they meet with your approval. We revised the paper according to your specific comments. Detailed explanations for the comments are shown below in the PDF files. 

Reviewer 2 Report

This study entitled ( Efficient Biorefinery Based on Designed Lignocellulosic Sub-2 strate for Lactic Acid Production ) successfully demonstrated the feasibility of fermenting brewer's spent grain (BSG), using E. mundtii. They optimized the hydrolysis of BSG with Celluclast 1.5L at an enzyme loading of 5 FPU/g biomass, generating pentose and hexose at a preferable ratio. The simultaneous saccharification and fermentation (SSF) process improved the utilization of fermentable sugars in fermentation, resulting in a maximum productivity of 3.65 g/L/h and the production of 53.1 g/L optically pure L-lactic acid. By controlling the pH at 7.0 during the SSF, they successfully minimized glucose-induced carbon catabolite repression (CCR). The lactic acid-producing strain was able to grow in BSG hydrolysate without the need for nutrient supplementation, as BSG provided both carbon and nitrogen sources. This study highlights the potential of using state-of-the-art fermentation techniques for efficient lactic acid production from lignocellulosic substrates, such as BSG, and establishes an efficient biorefinery approach for this process

The abstract effectively captures the main points of the study and its potential significance in the field.

The introduction provides a clear and comprehensive overview of the background and context of the study.

Materials and methods are appropriate for addressing the research questions and objectives.

In section 2.3. line 141 in the composition of MRS medium, do you mean Beef extract?

The results and discussion section presents the findings in a clear and organized manner, using appropriate figures, tables, and statistical analyses and the discussion effectively interprets the results and compares them with relevant literature.

The reference list is comprehensive and includes relevant and up-to-date sources.

Author Response

Thank you very much for reviewing our manuscript, your agreement, and valuable comments. We are also grateful for the favorable comments. We made corrections, and we hope they meet with your approval. Detailed explanations for the comments are shown below in the PDF files.

Reviewer 3 Report

Dear Authors,

The subject matter taken up by the authors extends the knowledge about the possibility of using lignocellulosic waste - brewer's spent grain (BSG) the barley malt residue generated from the brewing industry. The experiment was described correctly and in detail. Also, the description of the analytical method used - HPLC contains all the relevant information: chromatograph model, type and temperature of the chromatographic column, type and flow of the mobile phase, dosed sample volume. I specifically emphasize it, because often this data is missing and the reviewer must remind the authors to complete such information.

 I am only asking for clarifications and additions in minor matters.

1.      Chapter 2.2. Enzymatic hydrolysis

line 122: Why were BSG slurries with 10% solid content prepared for the study?

line 122: Why was Cellic CTec2 enzyme preparation chosen and not Cellic HTec2? Both preparations contain a complex of enzymes necessary for complete saccharification of complex sugars contained in the raw material.

line 124: I suggest adding that 1M Sodium acetate acted as a buffer.

line 126: It's "under 50°C". Please specify precisely, because under 50°C can mean, for example, 35 or 40 or 48/49°C.

2.      When cellulolytic enzymes Celluclast 1.5L and Cellic CTec2 are used for the hydrolysis reaction, do by-products form that may have a negative effect on the activity of enzymes or microorganisms in the subsequent lactic acid fermentation process?

I rate the manuscript highly.

Best regards,

Reviewer

Author Response

Thank you very much for reviewing this manuscript and giving me valuable comments. According to your comments, we revised the manuscript. Detailed modifications for the comments are shown below in the PDF files.
